# Importance of mobile genetic elements for dissemination of antimicrobial resistance in metagenomic sewage samples across the world

**Markus H. K. Johansson \*, Frank M. Aarestrup, Thomas N. Petersen**

National Food Institute, Technical University of Denmark, Kongens Lyngby, Denmark

\* markjo@food.dtu.dk

## Abstract

We are facing an ever-growing threat from increasing antimicrobial resistance (AMR) in bacteria. To mitigate this, we need a better understanding of the global spread of antimicrobial resistance genes (ARGs). ARGs are often spread among bacteria by horizontal gene transfer facilitated by mobile genetic elements (MGE). Here we use a dataset consisting of 677 metagenomic sequenced sewage samples from 97 countries or regions to study how MGEs are geographically distributed and how they disseminate ARGs worldwide. The ARGs, MGEs, and bacterial abundance were calculated by reference-based read mapping. We found systematic differences in the abundance of MGEs and ARGs, where some elements were prevalent on all continents while others had higher abundance in separate geographic areas. Different MGEs tended to be localized to temperate or tropical climate zones, while different ARGs tended to separate according to continents. This suggests that the climate is an important factor influencing the local flora of MGEs. MGEs were also found to be more geographically confined than ARGs. We identified several integrated MGEs whose abundance correlated with the abundance of ARGs and bacterial genera, indicating the ability to mobilize and disseminate these genes. Some MGEs seemed to be more able to mobilize ARGs and spread to more bacterial species. The host ranges of MGEs seemed to differ between elements, where most were associated with bacteria of the same family. We believe that our method could be used to investigate the population dynamics of MGEs in complex bacterial populations.

## Introduction

The increasing prevalence of bacteria with extensive antimicrobial resistance (AMR) is recognized as a significant threat to global public health [1] and estimates suggest that 1.3 million deaths annually can be attributed to AMR [2]. AMR can be acquired through either point mutations or by obtaining antimicrobial resistance genes (ARGs) via horizontal gene transfer [3]. This enables bacteria to exchange genetic information rapidly and, thus, confers a great

**Data Availability Statement:** All relevant data are within the paper and its Supporting Information files.

**Funding:** MHKJ, TNP, FMA recived funding from Novo Nordisk Foundation (www.novonordiskfonden.dk)(Grant: NNF16OC0021856: Global Surveillance of Antimicrobial Resistance) and the European Union's Horizon 2020 research and innovation programme (Grant: 874735). The funders had no role in study design, data collection and analysis, decision to publish, or preparation of the manuscript.

**Competing interests:** The authors have declared that no competing interests exist.

ability to adapt to environmental changes. Mobile Genetic Elements (MGEs) are discrete regions of DNA that can promote their own movement, or the movement of other MGEs, within or between bacterial cells. They are highly diverse and are divided into types based on their properties and genetic layout [4]. MGEs are fundamental for enabling this transmission of genes as they can recruit so-called accessory genes from the host and transpose themselves with the genes as a unit [4–6]. For that reason, MGEs are very important for bacterial evolution.

Intercellular transposing elements such as plasmids, integrative and conjugative elements (ICEs) integrative and mobilizable elements (IMEs), and cis-mobilizable elements (CIMEs) excel at spreading genetic material between bacteria as they can conjugate or be mobilized by the conjugation of other elements [6–8]. They frequently carry intracellular transposing MGEs, that only transpose between DNA within the same cell, thus enabling these MGEs to spread to new hosts. ICEs,IMEs and CIMEs can integrate into the host chromosome and are therefore said to be integrating MGEs while plasmids exist in the cytosol.

Intracellular transposing elements like Unit-transposons (Tn) and Composite transposons (ComTn) are integrated MGEs that carry accessory genes and are often associated with ARGs [4]. Some Tns are known for carrying integrons, a type of MGE capable of harboring and rapidly capturing new genes [9]. Insertion sequences (IS) are small intercellular transposing MGEs generally consisting of a transposase gene bounded by inverted repeats (IR). These elements can alter the gene expression either via gene inactivation [10,11] or by carrying outward-facing promoters [12]. While unable to carry accessory genes, IS elements can mediate the mobilization of nearby genes through the formation of ComTn [13,14]. Miniature inverted-repeats transposable elements (MITEs) are derivate of Tns or IS that consist of a pair of IR without a transposase and are therefore unable to self-transpose. Genes can be recruited and spread through this interplay of different types of MGEs [5,8]. Despite the importance of MGEs for spreading ARGs, little is known about the global distribution of these elements.

Sewage has been identified as an important factor for the accumulation and dissemination of ARGs as it can rapidly transport bacteria while acting as a reservoir [15]. As sewage has been shown to reflect the human microbiome, it has been suggested as a method for surveying changes in ARGs within and between a geographic area [16,17]. Previous studies have shown that ARGs in sewage cluster according to geographical regions, but whether this is the case for MGEs and whether there is any correlation between MGEs and ARGs has not been investigated [17,18].

Here we investigate the composition of MGEs and ARGs through 677 metagenomic sewage samples collected from 97 countries or regions worldwide. Our study aims to characterize MGEs present in human-associated bacteria, describe how they differ between geographical regions, and how they are associated with ARGs. Our study is the first to describe the prevalence of MGEs and their relation to ARGs at a global scale.

## Material and methods

### Dataset description and read processing

This study was conducted using previously published metagenomic data from the global sewage project from which we compiled a collection of 677 sewage samples [17]. The samples were collected from sewage plants in 97 countries and regions, representing all six continents, between 2016–2019. Only samples sequenced on the NovaSeq 6000 platform were included to avoid bias introduced by differences in sequencing technology. See S1 Appendix for sample information and ENA accession numbers. After read processing, the samples contained an average of 72 million reads (10.2 gigabases), of which 4.33% of the reads were mapped to MGEs and 0.05% to

AMR genes. Reads mapped to 3,850 different MGEs of the types included in this study, i.e., MITEs, IS, Tn, comTn, and conjugating transposons such as ICE, IME, and CIMEs.

## Estimation of MGE, AMR, and bacterial abundance

The abundance of MGEs, ARGs, and bacterial taxa was estimated by mapping reads to the ResFinder [19] database, the MobileElementFinder [20] database MGEdb (version 1.0.2), and the ribosomal typing database Silva [21] (version 38, downloaded 2020-01-16) with KMA [22] version 1.2.23. The Silva database contained at the time of mapping 2,225,272 16S rRNA sequences. Estimating abundances from 16S instead of from a non-redundant genomic database avoids the ambiguity of accessory genes shared by multiple species and has been successful in previous studies in estimating bacterial abundances [18,23]. MGE accessory genes were masked out from MGEdb to remove inter-dependency between the databases used. Acquired antimicrobial-, disinfectant- and heavy metal resistance genes and virulence factors carried in MGEs were identified with ResFinder and VirulenceFinder using the default thresholds for sequence identity and alignment coverage [24]. Reads were mapped with KMA using the options in S1 Table [22]. KMA uses the ConClave algorithm to assign reads to the most likely reference sequence in case there are multiple reference sequences with identical mapping scores. KMA was used to estimate all abundances to keep the bias introduced by the estimation algorithm the same for ARGs, MGEs, and bacteria. Sequences with less than ten read fragments mapping to them were not included to reduce the effect of spurious mapping and sequencing errors.

Due to DNA sequence data being compositional, the absolute abundance of the features can't be known, only the relative abundance of select features [25]. In addition, compositional data often needs to be normalized to have numerical properties that fulfill the assumptions of common statistical methods. In the analysis, we primarily relied on two types of log-ratio normalization fragments per kilobase reference per million bacterial fragments (FPKM) and centered log ratio (CLR) [26]. FPKM values are a version of the additive log ratio (ALR) [27] that sets the mapped fragment in relation to the bacterial content and feature length instead of a component of the composition. FPKM values were calculated according to Eq 1, where the number of fragments mapping to bacterial 16S was used to estimate the bacterial abundance.

$$FPKM = \frac{fragments\ mapping\ to\ a\ feature}{feature\ length \times bacterial\ abundance} \times 10^9 \tag{1}$$

Zeroes were replaced prior to log transformation using the Bayesian inference function in PyCoDa Python module [28]. The Shannon diversity index was used to describe the diversity of MGEs and ARGs in each sample [29].

## Hierarchical clustering of MGEs and ARGs

In a previous analysis of the same data, we found by visual inspection of PCoA plots that ARGs clustered according to geographical regions [18]. Here we studied the geographical distribution of MGEs and ARGs by clustering the samples on the CLR-transformed abundance profiles of ARGs and MGEs and using Ward distance. The number of clusters was determined from the resulting dendrograms (S1 and S2 Figs). Clustering was conducted using Scikit-learn (version 1.0.1) [30] and the geographical distribution of clusters was visualized using Plotly (version 5.5.6) [31] with shapefiles from Natural Earth. We used Scikit-bio (version 0.5.8) implementation of the Mantle test [32] to compare the degree of similarity between the ARG and MGE clusters. The Mantle test was used to calculate the two-sided Spearman correlation using 1000 random permutations of the ARG and MGE distance matrices.

## Statistical analysis and data visualization

We used the R package ALDEx2 [33] version 1.30.0 to conduct a differential abundance analysis. We aimed to identify types of MGEs and classes of ARGs that were more abundant in certain clusters and geographical regions. ALDEx2 was used to test for significant differences in CLR transformed abundances between sample groups using a Welch's t-test. The *P* values were corrected for multiple tests with the Benjamin-Hochberg false discovery rate (FDR) method [34]. We used 128 Dirichlet Monte-Carlo instances for estimating the posterior distribution for replacing zeroes prior to CLR transformation. The differential abundance was represented in an effect plot [35] which displays the within- and between-group variation in CLR values. Features with an FDR < 0.05 were reported as significant for the corresponding cluster. Graphics visualizing abundance, effect plot and the number of differentially abundant genes were generated with Python version 3.8.12 in conjunction with the Matplotlib (3.6.0) and Seaborn (0.12.1) modules.

## Identifying MGEs correlating with ARGs and bacterial genera

The tool fastspar, an implementation of the SparCC algorithm, was used to calculate the correlation between MGEs and ARGs and MGEs and bacterial genera [36,37]. Due to computational limitations, the number of features had to be reduced. CD-HIT was used to homology reduce the ResFinder collection of ARGs by clustering gene with 80% sequence identity and extracting the representative sequence for each cluster [38]. Each cluster is named by a representative sequence, i.e., the longest sequence in the group. The threshold was selected as it significantly reduced the number of features by grouping close homologs together within the same gene family. The ARG abundance was amalgamated based on these clusters and named after the representative sequence. MGEs were reduced by grouping them on their families when that information was available. When the MGE family was unavailable, the MGEs were reduced using the same CD-HIT homology reduction described above. Groups were named after the representative sequence or MGE family. A different methodology was used for MGEs, as elements of the same family can have rearrangements in their backbone or vary greatly in size which would not be taken into account by only reducing on nucleotide identity. For a full description of the groups, see S2 Appendix. This resulted in 623 ARGs and MGE groups, which were used as input to fastspar. To study how MGE abundance related to bacterial abundance, we used the homology-reduced set of MGEs and the abundance of bacterial phyla as input.

Correlations between features were calculated using 50 iterations and 20 exclusion iterations of highly correlated features. The reliability of the inferred correlations was calculated with the built-in bootstrapping method using 1,000 random permutations of the input. Correlations are assigned a p-value that reflects the probability that a more extreme correlation is observed in the permutations. Correlations with a correlation coefficient greater than 0.6 and a p-value lower than 0.05 were considered significant.

The phenotypic confinement of MGEs was calculated from the resulting correlations between MGEs and bacterial genera. Visualizations were conducted using a combination of Matplotlib and Seaborn [39,40].

## Results

### Composition and diversity of MGEs and ARGs

The relative abundance of MGEs, ARGs, and bacteria in the samples was estimated from the number of read fragments mapping to these elements. On average, 4.33% of the reads in the samples were assigned to MGEs and 0.05% to ARGs. There were hits to 3,850 different MGE, 1,682 ARG alleles, and reads mapped to 2,656 bacterial genera.

Intercellular transposing MGEs were the most abundant, constituting on average 96% of the FPKM normalized read counts. These MGEs are much smaller in terms of base pairs than conjugating MGEs and thus are, to a more considerable degree, impacted by the FPKM normalization as it takes bacterial content and feature length into account.

IS was the most abundant intercellular transposing MGE type, averaging 86,1% of all MGEs in a sample. IS are also the most common type of MGE in the reference database and one of the smallest types of MGEs in this study. The relative abundance of MGE types was roughly uniform across continents and exhibited less variation than the abundance of ARGs grouped on antimicrobial classes (S3 Fig). The ComTn type of MGEs tended to be in greater abundance in samples from Asia, Europe, and North America, while samples from Oceania had a higher abundance of Tn. Samples from Oceanian also had a greater abundance of macrolide and β-lactam resistance compared to Europe, which had a higher abundance of glycopeptide resistance. The diversity of MGEs and ARGs was comparable across the continents (Shannon diversity index 7.1 for MGEs and 5.8 for ARGs). One sample originating from Bangladesh had low ARG diversity (2.3) caused by 77.3% of the resistance being assigned to the macrolide resistance genes *msr(E)* and *mph(E)*.

## Geographical distribution of MGEs and ARGs

We aimed to identify how MGEs and ARGs are geographically distributed and if there are regional differences in the prevalence of certain genes or MGEs. Since ARGs are frequently mobilized by MGEs, we hypothesized that the distributions should be very similar. The samples were clustered based on their ARG and MGE abundance profiles. We determined four MGE and ARG clusters based on the dendrograms (S1 and S2 Figs). The sample clusters spanned multiple continents, showing that the distribution of these genes and MGEs is not limited to individual countries or continents.

Furthermore, the clusters had different geographical distributions, where some consisted of samples from all continents while others primarily contained samples from a few geographical regions (Fig 1). We designated the clusters as global or regional based on their geographical distribution. The MGE clusters 1 and 2, and ARG clusters 1, 2, and 4 were classified as regional. MGE cluster 4 and ARG cluster 3 were considered global as they contained several samples from multiple continents (Figs 1 and 2).

When investigating what differentiated the clusters, we found average ~1,000 MGEs with a significantly (*P*-value < 0.05) different abundance in these groups. However, for a majority of the MGEs the effect size was small, indicating that the difference might be of limited biological relevance (S4 Fig). Cluster 3 contained a higher abundance of MGEs with effect size > 0.5 for all MGE types (Fig 3A). All the 13 MGEs with a moderate effect (effect > 1) were found in cluster 3 and were primarily IS (8st) or ICEs (2st). Cluster 2 and 4 contained a greater number of MGEs with a significantly lower abundance than the other clusters (Fig 3C).

Regional MGE clusters seemed to separate according to the climate zones, as cluster 1 was primarily located in the tropical and subtropical zone. In contrast, the samples in cluster 2 were primarily found in the temperate zone. The Shannon information content was used to quantify how well the samples were ordered by climate zone per cluster. A bit score approaching zero means that samples are ordered by climate zone, while a score approaching one shows that samples are evenly distributed across the zones. The local MGE clusters 1 and 2, with a bit score of ~0.3, showed to be highly confined to either the temperate or subtropical/ tropical zones (S2 Table). The MGE clusters we classified as global scored ~0.59 and ~0.93 showing that they were not confined by climate zone.

The samples did not seem to separate according to climate zone when clustering on the ARG profile Instead, they tended to cluster according to continents, whereas samples from

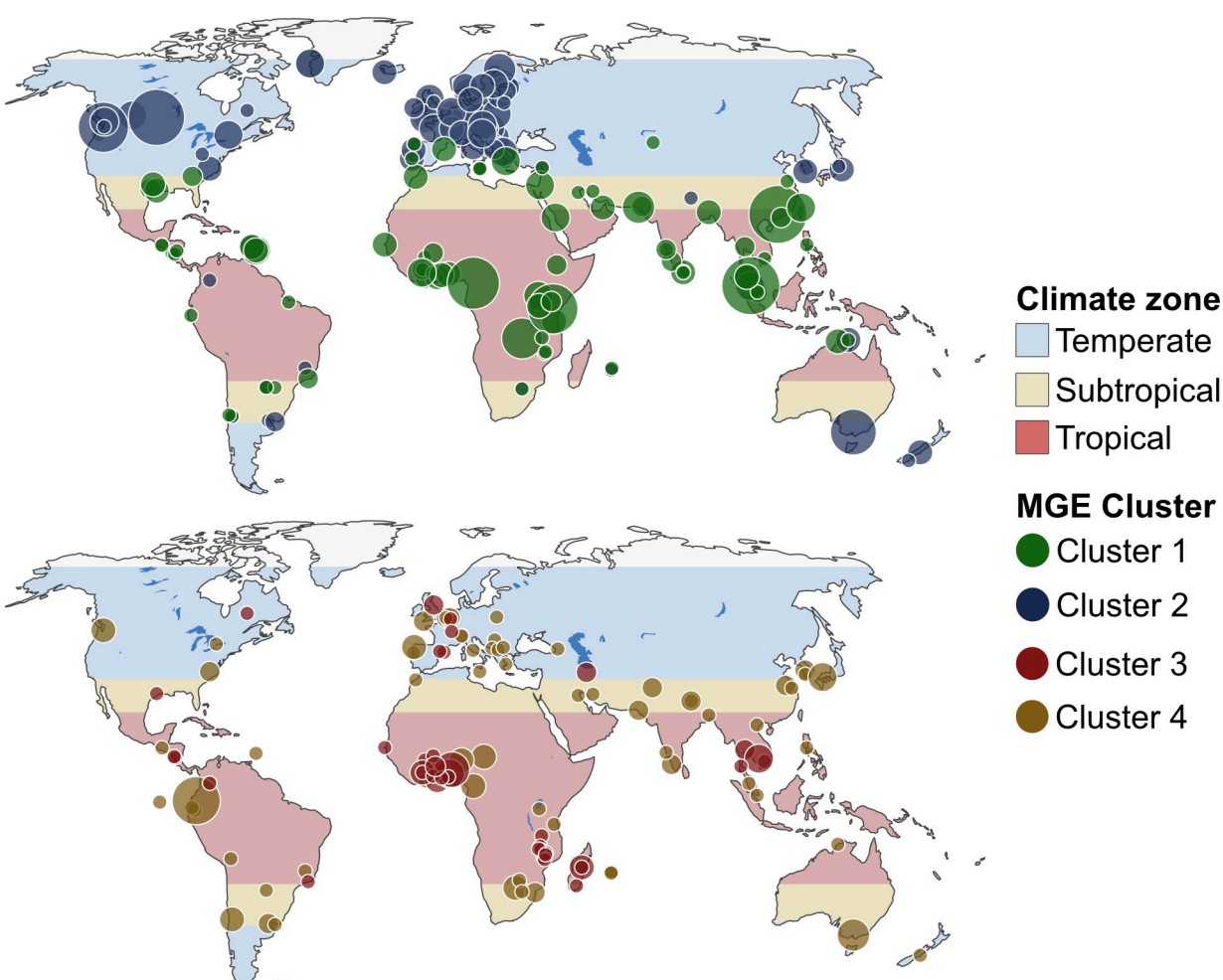

**Fig 1. Geographical distribution of samples clustered based on their relative MGE abundance profile.** The maps are colored by the Climate zone. Clusters 1, 2 and 3, 4 are drawn on separate maps.

Africa/Asia and Europe/America tended to cluster together. Despite ARG clusters 1 and 4 being comprised of samples primarily from the subtropics/tropics (bit score ~0.1, 0) we deemed them to separate according to continents as the samples they were comprised of were not evenly distributed along the equator like the MGE clusters (S2 Table and Fig 2). The difference in geographical separation patterns for MGEs and ARGs was further emphasized by ARG and MGE profiles to be uncorrelated (Spearman correlation coef ~0.61; p-value 0.001).

We found on average ~160 ARGs per cluster that had a significantly different abundance than the other clusters. The global ARG cluster 3 contained a greater number of significantly abundant resistance genes (effect > 0.5) than the other clusters (Fig 3B). The Africa-Asia dominant cluster 1 had higher abundance of five β-lactamase genes from the $bla_{OXA}$ family, three aminoglycoside genes, two lincosamide, and one macrolide gene. Cluster 2 contained a higher abundance of the folate pathway antagonist than the other clusters of which genes of the $sul$ and $dfrA$ families were the most prolifient. The central Africa dominated cluster 4 had higher abundance of the lincosamide resistance gene $lnu$(c) and three genes from $bla_{OXA}$ family. However, like MGEs were the effect size small for a majority of the resistance genes (S5 Fig). All significantly abundant ARGs can be found in S3 Appendix.

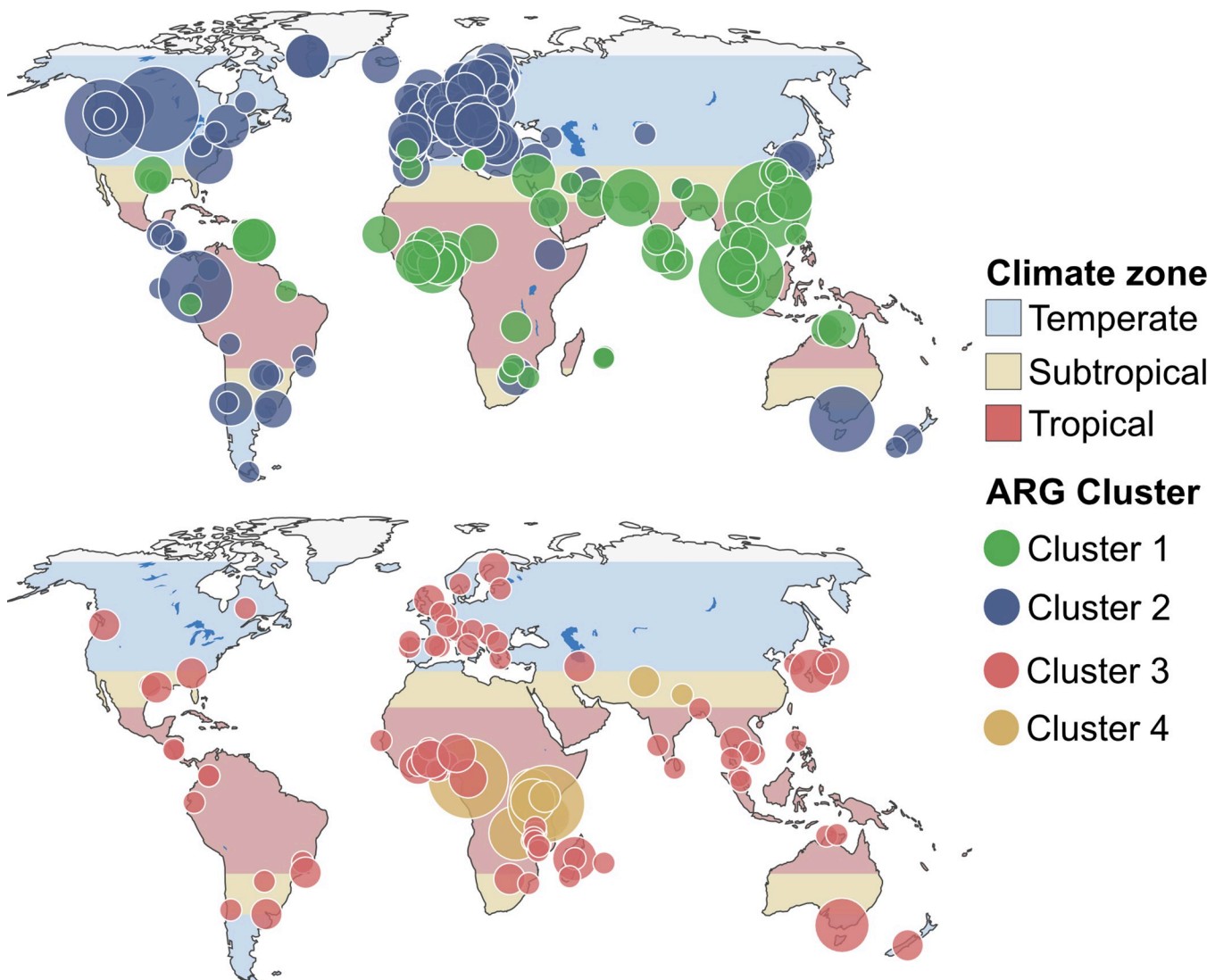

**Fig 2. Geographical distribution of samples clustered based on the abundance of ARGs.** Cluster 1 and 2 are shown on the upper map and cluster 3 and 4 are shown on the bottom map. The maps are colored according to climate zone.

## Correlation between MGEs and bacterial genera

We investigated the relationship between bacterial population and individual MGEs by identifying genera and MGEs whose co-abundance had a significant covariation. A significant covariation could indicate that the bacteria could act as a host for the MGE. The fastspar [37] implementation of the SparCC algorithm was used to calculate the correlation using the abundance of homology-reduced MGEs and genera as input. The bacterial abundance was estimated by read mapping to the Silva 16S database. All group members of the MGE clusters are detailed in S2 Appendix.

We identified 93 MGE groups whose abundance significantly correlated with the abundance of one or more bacterial genera. Each MGE group correlated with, on average, 4.8 different genera (std 4.7) (S4 Appendix). MGEs are primarily associated with the phyla Proteobacteria, Firmicutes, Actinobacteria, and Bacteriodetes (Fig 4A). These four phyla were also the most abundant taxa in the samples (S6 Fig).

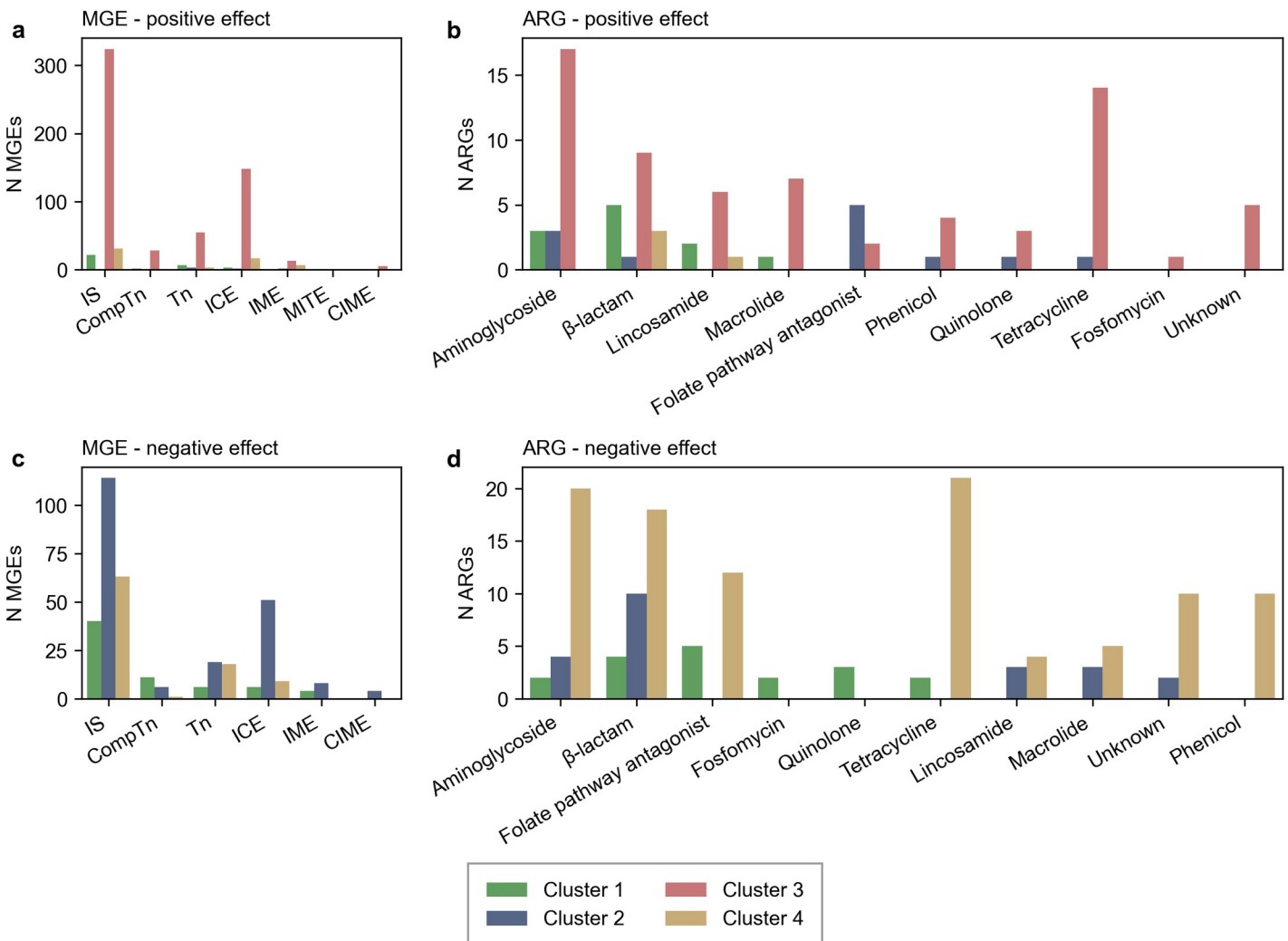

**Fig 3. Number of ARGs or MGEs with significantly different abundance per cluster.** Only features with an effect size greater than 0.5 was included. Figure a, b summarize the number of features with a positive effect size, figure c and d summarize the number of features with a negative effect size.

MGEs seemed to vary in their phylogenetic reach, with ~38.7% of MGEs being only associated with genera of the same family, ~47.3% with genera of the same order, and ~78.5% being associated with the same phylum (Fig 4B). Of the 21.5% of the MGEs associated with genera from different phyla were, the majority, intercellular transposing MGEs (ICE, IME, and CIME), which can carry accessory genes. Likewise, intercellular transposing MGEs constituted most of the MGEs that correlated with the most genera, highlighting the importance of ICE, IME, and CIME type MGEs for spreading genes throughout a bacterial population (Table 1).

MGEs that were associated with genera from multiple phyla tended to correlate with either gram- (Proteobacteria, Bacteriodetes, Verrucomicrobiota, and Fusobacteria) or gram+ (Actinobacteria and Firmicutes) phylum (S7 Fig). Some MGE groups did correlate with gram+ and gram- phyla; however, they were only associated with a single family outside the phyla they primarily associated with. For instance, Tn*916*, Tn*4453*, Tn*6103*, CTnBST, CIME other, and ICE-Sluban MGE groups primarily correlate with gram+ phyla and with a single family of the gram- Verrucomicrobiota. Likewise, were IS*6* family and Tn*6167* primarily associated with

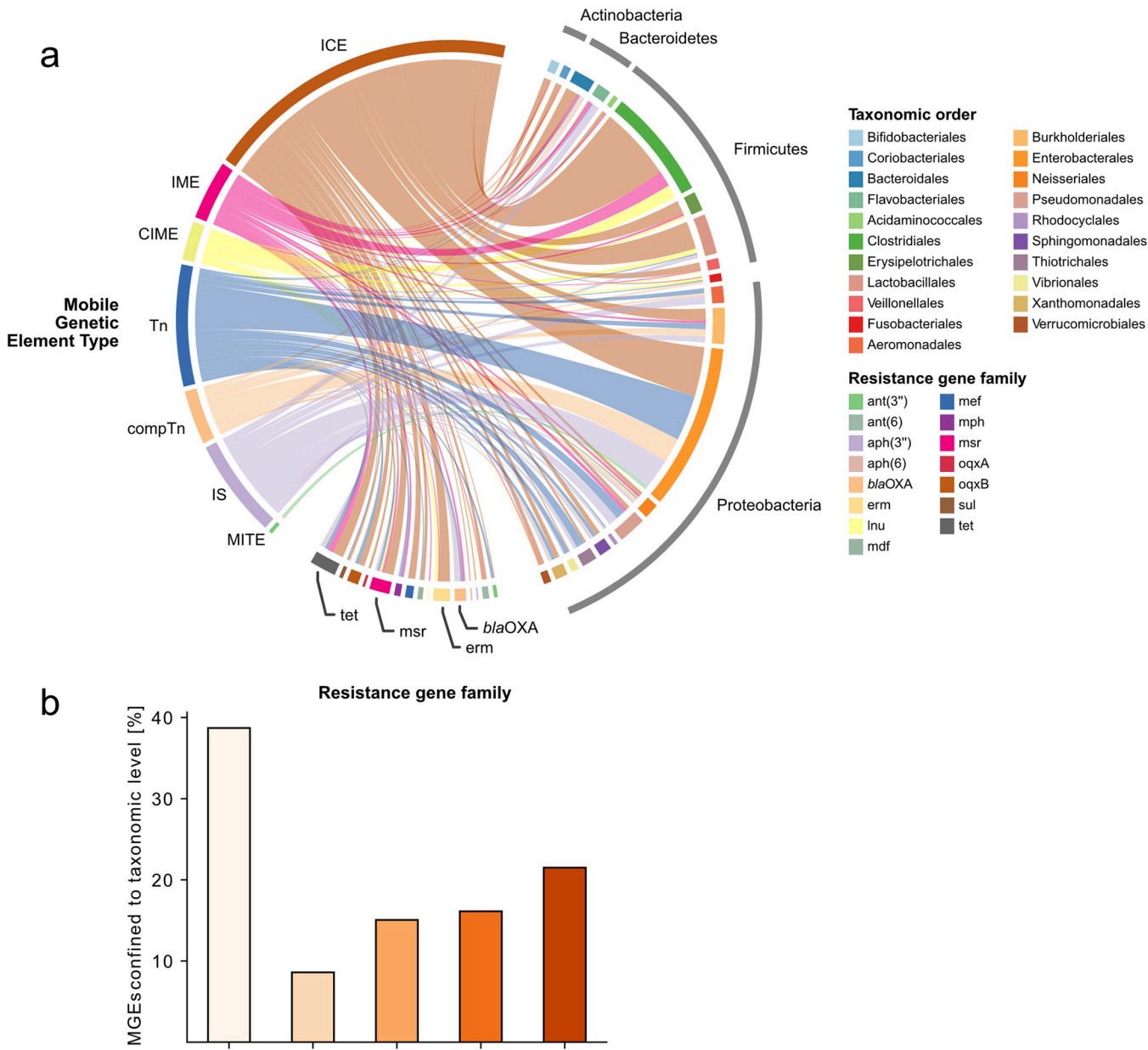

**Fig 4.** **a** Correlations between MGEs-ARGs and MGEs-genera. The chord width indicates the number of correlated features, a greater width corresponds to greater number of significant correlations. MGEs are amalgamated on type, ARGs on antibiotic class they confer resistance to and genera on phylum. **b Phylogenetic confinement of MGEs.** Average percentage of MGEs confided to a specific phylogenetic classification.

gram- but were also significantly correlated with one gram+ family of the Firmicutes phyla. This suggests that the host range of MGEs is limited by phylogeny and that gram- and gram + bacteria have different populations of MGEs.

Several of the MGEs found in multiple phyla also correlated with multiple ARGs, of which ant(6)-Ia, *erm*(B), and tet-like genes are the most common (Table 1 and 5S Appendix). This suggests that these MGEs have a greater phylogenetic reach and, thus, a greater potential to spread resistance genes to a broader set of bacterial taxa.

**Table 1. Highly disseminated MGEs based on the number of genera they correlated with.**

| MGE group[a] | MGE type | Number of genera | ARG groups[a] |
|---|---|---|---|
| **Tn916** | ICE | 24 | ant(6)-Ia, *erm*(B), *tet*(40), *tet*(O/32/O), *tet*(W) |
| **Tn6103** | ICE | 22 | ant(6)-Ia, *erm*(B), *tet*(O/32/O), *tet*(W) |
| **CTnBST** | ICE | 19 | ant(6)-Ia, *tet*(W) |
| **Tn4453** | IME | 19 | ant(6)-Ia, *erm*(B), *tet*(W) |
| **ICESluvan** | ICE | 18 | *erm*(B), *mef*(A), *msr*(D) |
| **CIME other** | CIME | 18 | ant(6)-Ia, *erm*(B) |
| **Tn4371** | ICE | 11 | 0 |
| **Tn6256** | ComTn | 10 | 0 |
| **Tn6285** | Tn | 8 | *oqx*B |
| **CTnHyb** | ICE | 8 | *cfx*A6, *tet*(Q) |

All MGEs correlated with several ARG groups. Groups are named after the representative sequence.

[a] at 80% sequence identity.

## Correlations between MGEs and ARGs

As various MGEs frequently mobilize ARGs, we investigated if certain MGEs are more prone to mobilize specific ARGs than others. The ARGs and MGEs were homology reduced to 80% sequence identity prior to calculating the abundance correlation. All group members of the MGE and ARG clusters are detailed in S2 Appendix.

We found 49 MGE groups whose abundance significantly correlated with the abundance of one or more ARG groups. Each MGE group correlated with, on average, two different groups of ARGs, with some correlating with up to five different ARGs (S4 Appendix). Of the ten MGE groups associated with the most ARG groups were seven intercellular transposing transposons, 2 Tns, and one insertion sequence (S3 Table). In addition, 4 of the MGE groups were inter-cellular transposing (ICE, CIME) which could enable the associated genes to be spread between bacteria without using another MGE like a plasmid. This indicates that they are more likely to mobilize different resistance genes or that the MGE and ARG genes are inherited as a conserved unit.

The intercellular transposing ICEs and IMEs and intracellular transposing Tns were, on average, associated with more ARG groups than other MGEs. They were primarily associated with macrolide, aminoglycoside, lincosamide, and tetracycline resistance (Fig 4A). The abundance of IMEs and Tns were also found to correlate stronger with β-lactam resistance genes of the $bla_{OXA}$-280 than other types of MGEs (IS and comTn), indicating that they are of importance for disseminating these genes (S8 Fig). Intercellular transposing MGEs (ICE, IME, and CIME) were also the only MGE types that significantly correlated with lincosamide resistance, specifically the *erm*(B) and *lnu*(C) genes.

## Correlations between ARGs

We also investigated which resistance gene groups significantly correlated with one another, as this could indicate the gene being inherited as a conserved unit. We found 23 ARG groups whose abundance significantly correlated with the abundance of one or more other ARGs. Interestingly, a few ARGs significantly paired up with specific combinations of up to 5 ARGs, i.e., correlated significantly. The macrolide resistance gene groups *msr*(E), *mph*(E) were strongly correlated (corr = 0.98) and also associated with the tetracycline resistance gene group *tet*(39) (corr = 0.63) (S9 Fig). One of the larger groups of correlated genes contained the β-lactamase gene $bla_{SHV}$-100, fosfomycin resistance *FosA6* and the *oqxA* and *oqxB* that yields

resistance to fluoroquinolones. We also found the gene groups aph(3"), aph(6), *sul*2 and *sul*1, ant(3"), *tet*(A) and *tet*(C) to be associated with one another.

## Discussion

The increasing prevalence of AMR bacteria is a significant concern for public health, as it increases the risk of infections and the cost of healthcare[1]. Many ARGs are known to have been mobilized by different MGEs, which has enabled them to spread rapidly[4,41]. Therefore, it is essential to understand how ARGs are mobilized, disseminated, and retained in human-associated bacterial populations. Previous studies have investigated the ability of MGEs to disseminate ARGs using reference genomes from sequencing repositories [42,43]. In contrast, our aim was to characterize the geographical distribution of MGEs and their ability to mobilize ARGs in the global bacterial population. To the best of our knowledge, this study is the first to examine the global geographical distribution and mobilizing potential of a broad set of integrated MGEs in human-associated metagenomics data.

Our analysis suggests that the MGE flora varies by geographical region. Some MGEs were found to be highly abundant in samples from all continents while other MGEs were primarily identified in samples from specific regions. Our analysis also showed that the regional variations of MGEs could be explained by samples originating from tropical or temperate zones. We speculate that climate or factors approximated by climate may affect the composition of MGEs, possibly by influencing the bacterial population which in turn affects MGEs. Interestingly, our previous observations that the ARG distribution tended to separate by continent were confirmed using hierarchical clustering[18]. The geographical distribution of ARGs was different from MGEs, suggesting that additional factors contribute to promoting the prevalence of ARGs. Further investigation of climate-induced effects on the MGE and ARG composition would require a more granular dataset with repeated sampling per location to account for seasons.

Fastspar was used to identify MGEs whose abundance correlated with the abundance of bacterial genera and ARGs. Fastspar uses a bootstrapping method to reduce false positives by estimating the likelihood of observing a more extreme correlation by randomly permutating the data [36]. MGE abundance was estimated using a database where accessory genes and nested MGEs had been masked out to reduce the risk of false positives caused by interdependencies.

We have identified 93 instances where a MGE significantly correlated with a bacterial genus. Our findings suggest that the host range of MGEs is limited by host phylogeny, as MGEs tend to be associated with phylogenetically related genera. This is consistent with previous studies that found phylogeny to be an important factor limiting the transposition pathways of ARGs[42,43]. When MGE groups were associated with genera from multiple phyla, the genera were mostly either all Gram- or Gram+, indicating that the MGE host range is limited by host phylogeny. However, some MGE groups, such as Tn*916*, Tn*4453*, and Tn*6167*, correlated with both Gram- and Gram+ bacteria, suggesting a greater ability to transfer genes between unrelated bacteria. However, further research is needed to verify the presence of these MGEs in the bacterial genera.

We have identified 49 MGE-ARG and 23 ARG-ARG pairs that have highly correlating co-abundances. These results suggest that these elements could be inherited together, either mobilized by the MGE or transported as part of a larger conserved unit, such as a plasmid. Several MGEs were associated with multiple ARGs, indicating that some MGEs have greater potential to disseminate a broader spectrum of genes. This could explain why the MGE and ARG profiles did not correlate. As expected, most of the MGEs associated with the greatest number of

ARGs were of types capable of carrying accessory genes or intercellular transposition, as these could directly transpose genes between multiple bacteria. Several of the ARG-ARG gene pairs have previously been described to be co-mobilized on plasmids, for instance *msr*(E), *mph*(E), and *tet*(39) in *Acinetobacter baumannii* [44,45] and aph(3"), aph(6), and *sul2* in *Escherichia coli* [46]. Although the combination of $bla_{SHV-100}$, *fos*A6, *oqx*A, and *oqx*B have not been described previously, are these genes prevalent in strains of *Klebsiella pneumoniae* from hospital wastewater [47] and are known to be mobilized by MGEs such as IS*26* [48,49].

There are several correlations between MGEs and ARGs that have not previously been reported, including the correlations between the Tn*6167*, Tn*6171*, and PGI1-PmPEL MGE groups and $bla_{OXA-280}$. The high number of novel associations is likely caused several factors. Many MGEs have only been reported a few times in the literature and often in a few clinically relevant genera not found in our data. For example, Tn*6167* has been described in four articles all investigating A. baumannii (PubMed search query "Tn6167"), Tn*6171* in one article (PubMed search query "Tn6171"), and IS*701* in six articles (PubMed search query "IS701"). Many Tns and conjugative transposons carry accessory genes in integrons, enabling them to rapidly exchange their accessory genes [9,50,51]. A recent study identified 13,397 integron-associated genes from environmental metagenomic samples, of which only 51 had previously been characterized [52]. It is likely that there are undescribed associations between resistance genes and MGEs.

Due to the complex nature of MGEs, co-occurrence cannot be inferred solely from correlating co-abundances. While there was support for some of the correlating features in the literature, further verification is needed. We attempted to use metagenome-assembled genomes (MAGs) to confirm co-mobilization, but poor yield of high-quality assemblies prevented the analysis. The high species diversity of sewage and insufficient sequencing depth probably caused the poor assemblies. Additionally, our attempts to use Oxford Nanopore sequencing during data generation only yield reads of insufficient length (approx. 2 Kbp).

Short-read sequencing was chosen due to its lower cost and higher throughput, which make it suitable for extensive studies that require deep sequencing to capture low-abundant organisms and genes. Sewage is a highly complex substrate containing DNA from multiple domains of life, including bacteria, protozoa, plants, and animals (including humans). A previous study using a subset of the dataset reported that on average 30% of the mapped reads aligned to bacterial genomes [17], highlighting the need for deep sequencing. Additionally, the completeness of the reference databases is a crucial factor in achieving high analytical sensitivity, underscoring the importance of continuous characterization and maintenance of MGE and ARG databases. We believe that short-read sequences are sufficient to capture differences in MGE populations and identify interesting relationships for future studies.

Our findings suggest that several factors influence the transposition network that drives the spread of ARGs. We found climate, or factors approximated by climate, to influence the MGEs present in a given geographic area, which in turn limits the MGEs available to bacteria and the potential MGE cross-interactions. We also found evidence that some MGEs have a broader host range than others, potentially making them more effective at spreading genes across different bacteria. These factors together limit the potential transposition pathways. We believe that our methodology could employed to investigate the population and dynamics of integrated MGEs in metagenomic samples.

## Supporting information

**S1 Fig. Hierarchical clustering of the samples on the CLR transformed MGE abundance using ward distance.** The four clusters are colored and named after the geographical

clustering in Fig *2*.
(TIF)

**S2 Fig. Hierarchical clustering of the samples on the CLR transformed ARG abundance using ward distance.** The four clusters are colored and named after the geographical clustering in Fig *2*.
(TIF)

**S3 Fig. Relative FPKM normalized abundance of MGEs per continent and MGE type.** Per sample abudance of MGEs was closed to 100 to display changes in the relative abundance within and between continents. b Relative FPKM transformed abundance of ARGs per continent. ARG abundance are grouped on the antibiotic class the gene yeilds resistance to. ARGs without assiged antibiotic class was excluded.
(TIF)

**S4 Fig. Analysis of significant differences in MGE abundance for the geographical clusters in figure 2S.** The relation of between cluster difference and within cluster dispersion of CLR transformed MGE abundances. Diagonal line show effect size of 1. MGEs with significant differential abundance (Benjamin-Hochberg corrected P value < 0.05) are colored according to MGE type.
(TIF)

**S5 Fig. Analysis of significant differences in ARG abundance for the geographical clusters in figure 3S.** The relation of between cluster difference and within cluster dispersion of CLR transformed ARG abundances. Diagonal line show effect size of 1. MGEs with significant differential abundance (Benjamin-Hochberg corrected P value < 0.05) are colored according to the antibiotic the gene yield resistance to.
(TIF)

**S6 Fig. Difference in relative abundance of bacterial phyla per MGE cluster.** Bacterial abundance was estimated from the number of fragments mapping to 16S. Phyla with a relative frequency lower than 0.05% of all mapped was combined into the other category. Abundances are CLR transformed. Samples in cluster 1 and 2 has higher content of Firmicutes than cluster 3 and 4; cluster 2 has higher content of Fusobacteria and cluster 4 has higher content of Proteobacteria.
(TIF)

**S7 Fig. Heatmap of the average correlation strength of MGEs and genera from different phyla and grouped by taxonomic family.** MGEs are colored on the type and taxonomic families are colored according to their phyla. Correlations was calculated on the relative abundance of homology reduced MGEs and only significant correlations was included. The rows were clustered using average linkage to display MGEs spanning multiple phyla.
(TIF)

**S8 Fig. Distribution of MGE-ARG correlation coefficient per antibiotic class and MGE type. ARGs that have not been assigned an AMR class in the ResFinder database are categorized as being of unknown class.**
(TIF)

**S9 Fig. Heatmap displaying the correlation strength between ARG groups.** Correlations was calculated on the relative abundance of homology reduced genes and only significant correlations was included. ARG groups were clustered using average linkage.
(TIF)

**S1 Table. Options used when mapping reads to the individual databases with KMA.**
(DOCX)

**S2 Table. Number of samples per cluster that are from the frigid/ temperate and subtropic/ tropic climate zone.** Shannon information was calculated from these observations and quantifies the randomness of the distribution where 0 is not random and 1 is evenly distributed.
(DOCX)

**S3 Table. The 10 MGE groups that correlated with the most different ARGs groups.** MGE groups are named after the representative MGE or MGE family.
(DOCX)

**S1 Appendix. Sample metadata including accession numbers.**
(XLSX)

**S2 Appendix. Homology reduced ARG and MGE groups.**
(XLSX)

**S3 Appendix. ARGs found to be significantly abundant per ARG cluster.**
(XLSX)

**S4 Appendix. MGEs whose abundance significantly correlated with bacterial genera.**
(XLSX)

**S5 Appendix. MGEs whose abundance significantly correlated with ARG abundance.**
(XLSX)

## Author Contributions

**Conceptualization:** Frank M. Aarestrup, Thomas N. Petersen.

**Data curation:** Markus H. K. Johansson.

**Formal analysis:** Markus H. K. Johansson.

**Funding acquisition:** Frank M. Aarestrup.

**Investigation:** Markus H. K. Johansson.

**Methodology:** Markus H. K. Johansson.

**Resources:** Markus H. K. Johansson.

**Software:** Markus H. K. Johansson.

**Supervision:** Frank M. Aarestrup, Thomas N. Petersen.

**Validation:** Markus H. K. Johansson.

**Visualization:** Markus H. K. Johansson.

**Writing – original draft:** Markus H. K. Johansson.

**Writing – review & editing:** Markus H. K. Johansson, Frank M. Aarestrup, Thomas N. Petersen.

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
