## [Decision Letter · Decision Letter 0]

4 Sep 2023

PONE-D-23-15400Importance of mobile genetic elements for dissemination of antimicrobial resistance in metagenomic sewage samples across the worldPLOS ONE

Dear Dr. Johansson,

Thank you for submitting your manuscript to PLOS ONE. After careful consideration, we feel that it has merit but does not fully meet PLOS ONE’s publication criteria as it currently stands. Therefore, we invite you to submit a revised version of the manuscript that addresses the points raised during the review process.

We look forward to receiving your revised manuscript.

Kind regards,

Mabel Kamweli Aworh, DVM, MPH, PhD. FCVSN

Academic Editor

PLOS ONE

Journal Requirements:

3. We note that Figures 1 & 2 in your submission contain [map/satellite] images which may be copyrighted. All PLOS content is published under the Creative Commons Attribution License (CC BY 4.0), which means that the manuscript, images, and Supporting Information files will be freely available online, and any third party is permitted to access, download, copy, distribute, and use these materials in any way, even commercially, with proper attribution. For these reasons, we cannot publish previously copyrighted maps or satellite images created using proprietary data, such as Google software (Google Maps, Street View, and Earth). For more information, see our copyright guidelines: http://journals.plos.org/plosone/s/licenses-and-copyright.

a. You may seek permission from the original copyright holder of Figures 1 & 2 to publish the content specifically under the CC BY 4.0 license. 

Natural Earth (public domain): http://www.naturalearthdata.com/.

Additional Editor Comments (if provided):

In addition to addressing the comments raised by the reviewers kindly highlight the limitations of this present study

Reviewers' comments:

Reviewer's Responses to Questions

**Comments to the Author**

1. Is the manuscript technically sound, and do the data support the conclusions?

Reviewer #1: Yes

Reviewer #2: Yes

2. Has the statistical analysis been performed appropriately and rigorously? 

Reviewer #1: Yes

Reviewer #2: Yes

3. Have the authors made all data underlying the findings in their manuscript fully available?

Reviewer #1: Yes

Reviewer #2: Yes

4. Is the manuscript presented in an intelligible fashion and written in standard English?

Reviewer #1: Yes

Reviewer #2: Yes

5. Review Comments to the Author

Reviewer #1: In this study, the authors have analysed the dataset consisting of 677 metagenomic sequenced sewage samples from 97 countries or regions and studied how ARGs associated MGEs are geographically distributed worldwide. Overall, this study seems to be conceptual research with significant information about MGEs and ARGs. Below are some minor comments:

1. Provide expansion for abbreviation in the first mention (“ARGs” in abstract section)

2. Line 78 – Provide expansion for MITEs

3. Line 79 – Provide expansion for CIMEs

4. Kindly move the sentence “Insertion sequences had the highest relative abundance…….Africa exhibited the greatest variance in relative MGE abundance” to result section

5. Line 86-87 – Provide reference or website link for ResFinder, MGEdb and ribosomal typing database

6. How do these intercellular transposing mobile elements carry only particular resistance genes. Is there any mechanism of selection?

7. The authors explained that the regional differences in MGE flora is due to climate. Did the authors get a chance to correlate the prevalence of ARGs with antibiotic policy/treatment guidelines followed by each country included in this study?

8. Any information about free-floating extracellular DNA (exDNA) that are reported to carry a substantial amount of ARGs and MGEs in sewage?

9. Fig S8 – Define “unknown” either in the manuscript or in the figure legend.

10. Conclusion part needs little more clarity

Reviewer #2: 1. The manuscript is technically sound and relevant to addressing the burden of antimicrobial resistance. Much data has been shared by the authors including some supplementary data.

2. Statistical analysis was detailed with supporting material.

3. The authors made their data available including supplementary data 1-5.

4. The authors also made a good attempt in presenting the manuscript with a good flow chronologically.

5. In the abstract (line 11), the first usage of ARGs should be defined.

6. In lines 33 and 34, MGE and MGEs are used. I suggest the authors maintain the use of just MGEs.

7. In lines 34-35, “within/or between” may rather be “within/between” or “within or between”.

8. As noticed in lines 40 and 43, I suggest the authors use “IMEs” instead of “IME” since it is plural. Same may apply to “ICEs” instead of “ICE” as seen in line 39.

9. In line 79, “Insertion sequences” should be “IS” as it had been defined earlier in line 49.

10. In line 115, “Hierarchically” should be “Hierarchical”.

11. In line 128, “who” may rather be “’which”.

12. In line 175, the authors defined FPKM again after defining it earlier in line 105.

13. In line 326, “was” should be “were”.

14. The statement in line 343 may need to be reconstructed.

15. Kindly reconstruct the statements from lines 350-355 for clarity.

16. I suggest lines 395-396 should read as “For instance, msr(E), mph(E), and tet(39) genes have been found….”.

17. I suggest the statement in lines 428-429 be rephrased to “The transposition network is therefore influenced by the MGEs which are available in the population of bacteria ultimately”.

18. Most of the references are old (more than 5 years). Only about 19 out of 62 references are between 2018 and 2023. The authors should update the references with about 80% being within the last five years.

19. Generally, this is a very important manuscript in addressing antimicrobial resistance.

6. PLOS authors have the option to publish the peer review history of their article (what does this mean?). If published, this will include your full peer review and any attached files.

Reviewer #1: **Yes: **Dr. Dhiviya Prabaa Muthuirulandi Sethuvel

Reviewer #2: No

---

## [Author Response · Author response to Decision Letter 0]

14 Sep 2023

Reviewer 1

=======

Response to comment 1

Added definition of ARGs in the abstract

Response to comment 2

Added definition and a short description of MITEs to the introduction.

Added: “Miniature inverted-repeats transposable elements (MITEs) are derivate of Tns or IS that consist of a pair of IR without a transposase and are therefore unable to self-transpose.”

Response to comment 3

Added definition of CIMEs to the introduction.

Response to comment 4

The sentences were moved to the results section and incorporated in the text body

Response to comment 5

Added: Citations to ResfFinder, MobileElementFinder and SILVA database.

Response to comment 6

It is likely caused by multiple factors, some of which might still be unknown. Some factors likely contributing to stable ARG - intercellular transposing MGE (interMGE) combinations are described below.

interMGEs can receive new accessory genes through homologous recombination, insertion of another MGE carrying accessory genes, or carrying integrons. The host bacteria heavily repressed MGE transposition and integron activity, limiting the gene exchange for intercellular transposing MGEs. Several transposons also exhibit transposon immunity, meaning they can prevent the insertion of additional related transposons to "protect" themselves from being inactivated by transposing MGEs. Likewise, plasmid incompatibility prevents the accumulation of closely related plasmids.

Harboring interMGEs comes with a fitness cost to the host bacteria because conjugation and expression of accessory genes are costly processes. The fitness cost is reduced by tight regulation of conjugation and MGE gene expression. Integrons often harbor multiple resistance genes in their cassette arrays, enabling MGEs to carry more beneficial genes without inflating the fitness cost. Experiments by Wein et al. (2019)(2) demonstrated that by reducing fitness cost ARG-carrying plasmids tend to become fixed in the population under non-selective conditions. In addition, several plasmids have post-segregational killing systems that ensure the plasmid is inherited during host cell division.

Finally, several MGEs are known to be highly associated with specific ARGs. For instance, Tn917 carries ermA, Tn1721 - tetA, and Tn552 - blaZ. Recent studies(3, 4) (including this) suggest that the MGE host range is limited by host phylogeny. If true, this indicates that interMGEs are primarily exposed to a limited number of integrated MGEs and their accessory genes, thus acting as a stabilizing factor.

1. Wein, T., Hülter, N. F., Mizrahi, I., & Dagan, T. (2019). Emergence of plasmid stability under non-selective conditions maintains antibiotic resistance. Nature Communications, 10(1). https://doi.org/10.1038/S41467-019-10600-7

2. Hu, Y., Yang, X., Li, J., Lv, N., Liu, F., Wu, J., Lin, I. Y. C., Wu, N., Weimer, B. C., Gao, G. F., Liu, Y., & Zhu, B. (2016). The bacterial mobile resistome transfer network connecting the animal and human microbiomes. Applied and Environmental Microbiology, 82(22), 6672–6681. https://doi.org/10.1128/AEM.01802-16

3. Ellabaan, M. M. H., Munck, C., Porse, A., Imamovic, L., & Sommer, M. O. A. (2021). Forecasting the dissemination of antibiotic resistance genes across bacterial genomes. Nature Communications, 12(1), 1–10. https://doi.org/10.1038/s41467-021-22757-1

Response to comment 7

We chose not to investigate the correlation between ARGs and factors such as antimicrobial usage based on the findings of a previous study conducted by Hendriksen et al (2019)(1). They analyzed the global abundance of ARGs using this dataset, which consisted of 82 samples at the time. Using a random forest model and data from the European Centre for Disease Prevention and Control (ECDC), IQVIA, and the World Bank, they identified factors that correlated with high ARG abundance. Their findings indicated that resistance primarily correlated with sanitation and public health factors, and they did not find a significant correlation between antimicrobial usage (AMU) and resistance burden.

1. Hendriksen, R. S., Munk, P., Njage, P., van Bunnik, B., McNally, L., Lukjancenko, O., Röder, T., Nieuwenhuijse, D., Pedersen, S. K., Kjeldgaard, J., Kaas, R. S., Clausen, P. T. L. C., Vogt, J. K., Leekitcharoenphon, P., van de Schans, M. G. M., Zuidema, T., de Roda Husman, A. M., Rasmussen, S., Petersen, B., … Consortium, T. G. S. S. project. (2019). Global monitoring of antimicrobial resistance based on metagenomics analyses of urban sewage. Nature Communications, 10(1), 1124. https://doi.org/10.1038/s41467-019-08853-3

Response to comment 8

We did not investigate exDNA because of limitations with our data. We initially tried to construct metagenomic assembled genomes (MAGs) from the short-read sequenced data, but we struggled with a poor yield of high-quality genomes. We believe that this was due to the relatively low bacterial DNA concentration in combination with the many species (from all domains) contained in sewage. Additionally, our attempts to use Oxford Nanopore sequencing did not yield reads of sufficient length (approximately 2 Kbp).

The poor assembly quality would make it difficult to determine whether an MGE or ARG was located on a fragmented chromosome, plasmid, or free-floating DNA

.

Response to comment 9

We have described what Unknown AMR class are “S8 Fig” figure legend.

Added: “ARGs that have not been assigned an AMR class in the ResFinder database are categorized as being of unknown class.”

Response to comment 10

We have reworked discussion and conclusion to be more clear and to read better.

Reviewer 2

=======

Response to comment 1-4, 19

We thank you for you kind comments and are glad that you found merit in our work.

Response to comment 5

Added definition of ARGs in the abstract

Response to comment 6

Changed: MGE to MGEs.

Response to comment 7

Changed: within/ or between to within or between.

Response to comment 8

Changed: IME to IMEs

Changed: ICE to ICEs

Response to comment 9

This was corrected when the sentence was moved from the Materials and Methods section to the Results section. The sentence was reworked to fit in the body of text.

Response to comment 10

Changed: hierarchically to hierarchically.

Response to comment 11

Changed: who to that.

Response to comment 12

Removed: Redundant FPKM definition in results section.

Response to comment 13

Changed: was to were.

Response to comment 14

Changed: “Sewage is a highly complex material that contains among other bacterial, protozoa, plant and human DNA.” to “Seewage is a highly complex substrate that contains DNA from multiple domains of life, including bacteria, protozoa, plants, and animals (including humans).”

Response to comment 15

The paragraph has been rewritten to more clearly describe our choice in sequencing and analysis methods.

Changed: ”Analysis of MGEs is challenging in the sense that they are much longer than standard short-read Illumina sequences. One option was to assemble short read-sequences into longer contigs but we saw too many partial MGEs. Another option was to use long-read sequencing but here the output in base pairs was much lower compared to what we could obtain with Illumina short-read sequencing when the study was conducted, and the budget available to sequence worldwide metagenomics samples. Also, in our hands the attempt to obtain long DNA sequences for Oxford Nanopore sequencing did not yield sufficiently long reads (approx. 2 Kbp) at the time of the experiment. We therefore opted to use short-read sequences and perform a co-abundance analysis to identify pairwise co-abundance correlations knowing that correlation does not necessarily mean co-existence.”

to

“We used short-read sequencing to investigate the abundance of MGEs and ARGs. Short-read sequencing was chosen because of its lower cost and greater throughput than long-read platforms. At the time of the experiment, our initial attempts with Oxford Nanopore sequencing did not yield sufficiently long reads (approx. 2 Kbp).

The relationship between MGEs, ARGs, and genera was studied by analyzing co-abundances estimated from read mapping, knowing that correlation does not necessarily mean co-existence. This method was chosen because preliminary experiments with constructing metagenomic assembled genomes (MAGs) struggled with highly fragmented genomes. This was likely caused by low bacterial DNA concentration in combination with MGEs being much larger than the read length.”

Response to comment 16

We agree with your suggestion.

Changed: “For instance, have msr(E), mph(E), and tet(39) genes been found…” to “For instance, msr(E), mph(E), and tet(39) genes been found”

Response to comment 17

Changed: “The transposition network is therefore influenced by the which MGEs are available in the population of bacteria ultimately” to “The transposition network is therefore ultimately influenced by the MGEs which are available in the population of bacteria”

Response to comment 18

We understand and agree with your concern and have updated the references when possible and applicable.

It is important to note that evaluating references solely based on their publication year is not enough. In the material and methods section, we cite 24 articles or websites of which 14 were published before 2018. These older references relate to specific software, databases, foundational concepts, or methods used during the analysis that are relevant even if they are old. Databases and software such as Resfinder, Scikit-learn, and CD-HIT are continuously developed. Likewise, fundamental statistical and data normalization methods are still widely used and relevant. We have also cited some older manuscripts, such as "Microbiome Datasets Are Compositional: And This Is Not Optional" because it provides an excellent explanation of why compositional analysis must be used when analyzing microbiome data. Even though these manuscripts are old, their concepts still hold true.

The current version of the manuscript includes 52 references. Approximately 70% of the references, excluding those in the Materials and Methods section, were published in the last five years.

---

## [Decision Letter · Decision Letter 1]

9 Oct 2023

Importance of mobile genetic elements for dissemination of antimicrobial resistance in metagenomic sewage samples across the world

PONE-D-23-15400R1

Dear Dr. Johansson,

We’re pleased to inform you that your manuscript has been judged scientifically suitable for publication and will be formally accepted for publication once it meets all outstanding technical requirements.

Kind regards,

Mabel Kamweli Aworh, DVM, MPH, PhD. FCVSN

Academic Editor

PLOS ONE

Additional Editor Comments (optional):

Reviewers' comments:

Reviewer's Responses to Questions

**Comments to the Author**

1. If the authors have adequately addressed your comments raised in a previous round of review and you feel that this manuscript is now acceptable for publication, you may indicate that here to bypass the “Comments to the Author” section, enter your conflict of interest statement in the “Confidential to Editor” section, and submit your "Accept" recommendation.

Reviewer #1: All comments have been addressed

Reviewer #2: All comments have been addressed

2. Is the manuscript technically sound, and do the data support the conclusions?

Reviewer #1: Yes

Reviewer #2: Yes

3. Has the statistical analysis been performed appropriately and rigorously? 

Reviewer #1: Yes

Reviewer #2: Yes

4. Have the authors made all data underlying the findings in their manuscript fully available?

Reviewer #1: Yes

Reviewer #2: Yes

5. Is the manuscript presented in an intelligible fashion and written in standard English?

Reviewer #1: Yes

Reviewer #2: Yes

6. Review Comments to the Author

Reviewer #1: (No Response)

Reviewer #2: The authors have adequately responded to the initial review made and made significant improvement.

7. PLOS authors have the option to publish the peer review history of their article (what does this mean?). If published, this will include your full peer review and any attached files.

Reviewer #1: **Yes: **Dhiviya Prabaa Muthuirulandi Sethuvel

Reviewer #2: No

---

## [Editor Report · Acceptance letter]

11 Oct 2023

PONE-D-23-15400R1 

Importance of mobile genetic elements for dissemination of antimicrobial resistance in metagenomic sewage samples across the world 

Dear Dr. Johansson:

I'm pleased to inform you that your manuscript has been deemed suitable for publication in PLOS ONE. Congratulations! Your manuscript is now with our production department. 

Kind regards, 

on behalf of

Dr. Mabel Kamweli Aworh 

Academic Editor

PLOS ONE